# Application of Detrended Fluctuation Analysis and Yield Stability Index to Evaluate Near Infrared Spectra of Green and Roasted Coffee Samples

**Eszter Benes [1,2], Marietta Fodor [1], Sándor Kovács [3,*] and Attila Gere [2]**

[1] Institute of Food Quality, Safety and Nutrition, Szent István University, Villányi Str. 29-31, H-1118 Budapest, Hungary; eszter.benes@gmail.com (E.B.); Fodor.Marietta@etk.szie.hu (M.F.)

[2] Institute of Food Technology, Szent István University, Villányi Str. 29-31, H-1118 Budapest, Hungary; gereattilaphd@gmail.com

[3] Department of Economical and Financial Mathematics, University of Debrecen, Böszörményi út 138, H-4032 Debrecen, Hungary

[*] Correspondence: kovacs.sandor@econ.unideb.hu

**Abstract:** Coffee quality, and therefore its price, is determined by coffee species and varieties, geographic location, the method used to process green coffee beans, and particularly the care taken during coffee production. Determination of coffee quality is often done by the nondestructive and fast near infrared spectroscopy (NIRS), which provides a huge amount of data about the samples. NIRS data require sophisticated, multivariate data analysis methods, such as principal component analysis, or linear discriminant analysis. Since the obtained data are a set of spectra, they can also be analyzed by signal processing methods. In the present study, the applications of two novel methods, detrended fluctuation analysis (DFA) and yield stability index (YSI), is introduced on NIR spectra of different roasting levels of coffee samples. Fourteen green coffee samples from all over the world have been roasted on three different levels and their NIR spectra were analyzed. DFA successfully differentiated the green samples from the roasted ones, however, the joint analysis of all samples was not able to differentiate the roasting levels. On the other hand, DFA successfully differentiated the roasting levels on samples level, which was strengthened by a 100% accurate agglomerative hierarchical clustering. YSI was first used in NIR signal processing and was able to detect that a light roast is the most stable among all roasting levels. Future research should focus on the application of DFA in terms of the analysis of the effects of other transformation methods of the spectra.

**Keywords:** *Coffea arabica*; grouping; different roasting levels; yield stability index; detrended fluctuation analysis

## 1. Introduction

Coffee is one of the most marketed products worldwide. On a large scale, the quality of the beverage depends on the physical properties and chemical composition of raw coffee beans, respectively, on the roasting process. The quality of the coffee beverage is manifested differently depending on its geographic origin, and there are notable variations in sensory profile according to country of origin, microregions, and even different planting locations in the same farm [1].

Currently, over 100 species within the genus *Coffea* are catalogued. Despite this diversity, only two species are actually of great importance in the world market, *C. arabica* L. and *C. canephora* Pierre [1]. These two species have distinct characteristics, especially regarding the quality of the beverage. However, within the species *C. arabica*, there are also differences in the peculiarity of the varieties [2]. Coffee quality, and therefore price, is determined by coffee species and varieties,

geographic location, the method used to process green coffee beans, and particularly the care taken during coffee production [3,4]. To make difference between commodity and specialty coffee, different grading systems are used. Increasing awareness of quality, taste and health among consumers is increasing demand for high-quality and specialty coffees [5].

Chemical constituents of the roasted beans determine the quality of coffee as a beverage. Raw coffee beans contain a wide range of different chemical compounds, which react and interact amongst themselves at all stages of coffee roasting, resulting in greatly diverse final products [6]. Hence, certain coffees due to their chemical composition are better roasted to certain colors. In case of specialty coffees, lighter roasts have come to predominate. The roaster's responsibility is now phrased as revealing the character of the beans. While lighter roasts are trendy in specialty coffee, roasting too light, under-roasting, produces a thin, grassy flavor. Coffee assessment is usually undertaken when the bean is just past this stage but still fairly light. The beans are properly roasted and all the potential details of the coffee are apparent [6].

Near infrared spectroscopy (NIRS) is a nondestructive and noninvasive analytical technique, with minimal or no sample preparation. It is fast, low cost, robust, and can be used in different environments such as laboratories and industrial plants [3]. It is widely used in different fields of food industry for qualitative and quantitative analysis of products from raw material to finished products [7–9].

Several scientific studies have been published about the use of NIRS to investigate various properties of raw and roasted coffee beans, such as caffeine and chlorogenic acid content [10,11], color and defectiveness of beans [12]. Besides, NIRS can be used to analyze roasting conditions [13]; determine the roasting degree [14] and Arabica/Robusta ratio [15,16] in ground coffee; the place of origin of coffee beans [17,18]; chemical composition of coffee grounds [19] and the sensory properties of beverages [20–23]. However, in order to extract the most information from NIRS measurements, different data analyses methods are needed. In the aforementioned scientific papers, different pre-processing techniques (MSC, SNV, first and second derivatives, normalization, OSC) and multivariate statistical tools such as PCA, PLSR, linear discriminant analysis (LDA) and PLS-DA were used to analyze various properties of coffee.

The first step in NIR data analysis is the visual representation of the recorded signal responses on the recording wavelengths. The characteristics of the created curves might be able to explain differences among samples and highly different samples can be differentiated visually. The obtained NIR spectra, however, need to be further analyzed using multivariate statistical methods in order to extract more information to be able to define small differences among samples. Several different statistical methods have been introduced such as partial least squares discriminant analysis (PLS-DA) [17], soft independent modelling of class analogy (SIMCA) [24], principal component analysis (PCA), and locally weighted regression (LWR) [25], multivariate curve resolution-alternating least squares (MCR-ALS) method [26], and k-nearest neighbors and support vector machine (SVM) [27]. The majority of these methods treat NIR spectra as an observations–variables table, where the samples are considered as observations and the given wavelengths are used as variables. However, NIR spectra can also be considered as time series since the responses are recorded at equally distanced wavelengths.

One promising technique, detrended fluctuation analysis (DFA) has been introduced for the analysis of chromatograms [28]. DFA is a widely used time series data analysis tool and was successfully applied in the past few years for the evolution of high-viscosity gas–liquid flows [29], water contaminant classification [30], for EEG patterns associated with real and imaginary arm movements [31], air traffic flow analysis [32], and even for the analysis of NBA results [33], for instance. As the above list of completely different fields of application of DFA suggests, the method is highly flexible and can easily be integrated into different systems.

Another promising approach in NIRS data analysis would be the application of yield stability index (YSI) [34]. Agricultural production might provide high fluctuations due to environmental factors. However, there is always a clear yield range a farmer can handle. In most years, the yield fluctuates

within this range; however, there are years when extremely high or low yields might cause serious economic losses. YSI has been developed to measure these extremities in a time series by quantifying the level of stability for a yield series by measuring the proportion of annual yields being reasonably close to the expected trend value within a time period [35]. In NIRS data analysis, YSI is expected to provide information about the stability of the signals of the different roasting levels, e.g., which spectra shows the highest fluctuation.

In this study, the applications of detrended fluctuation analysis and yield stability index is introduced on NIR spectra of different roasting levels of coffee samples. The aims of the study are as follows:

−  Introduce new tools for the analysis of near infrared spectra;
−  Differentiate coffee samples based on their roasting levels using detrended fluctuation analysis.

## 2. Materials and Methods

### 2.1. Samples

Fourteen green *Coffea arabica* samples from different geographical origin and one *Coffea canephora* sample from India were purchased from Semiramis Kft. (Budapest, Hungary). Detailed information about the samples are shown in Table 1. These samples were divided in three part after recording their near infrared spectra to perform the roasting experiment.

**Table 1.** Description of coffee samples.

| Sample Number | Geographical Origin |
| --- | --- |
| 1 | Brazil, South America |
| 2 | India, Asia |
| 3 | Uganda, Africa |
| 4 | Colombia, South America |
| 5 | Uganda, Africa |
| 6 | Colombia, South America |
| 7 | Sumatra, Asia |
| 8 | Papua New Guinea, Asia |
| 9 | Guatemala, Central America |
| 10 | Kenya, Africa |
| 11 | Ethiopia, Africa |
| 12 | Panama, Central America |
| 13 | Mexico, Central America |
| 14 | India, Asia |
| 15 | Uganda, Africa |

### 2.2. Roasting Experiment

The samples were small scale roasted using a pre-determined roasting method. Three roasting levels were set up considering the first crack of samples: light, medium and dark roast. In the case of light roasted coffees, the samples were removed from the roaster 40 s after the first crack. To obtain the other two levels, the roasting time was raised to 60 s and 90 s after the first crack. The initial temperature of roasting was around 150 °C and the final temperature around 190 °C. From each raw sample, about 300 g were roasted to each level using a Probat roaster (Sample Coffee Roaster, Leogap, Curitiba, PR, Brazil).

### 2.3. Near-Infrared Spectroscopy (NIRS)

The near-infrared spectra of the green and roasted coffee beans were collected using a Bruker MPA™-Multipurpose FT-NIR analyzer (Bruker, Ettlingen, Germany). For the measurement, the whole amount of the raw and roasted samples was used, of which 40–60 g were measured using a rotating

cuvette (Ø85 mm) in six replicates. The spectral data were collected in diffuse reflection measurements mode within the range of 12,500–3800 cm$^{-1}$ (resolution 16 cm$^{-1}$; scanning speed 10 kHz), using the OPUS 7.2 (Bruker, Ettlingen, Germany) software. Each spectrum was calculated as the average of 32 subsequent scans. Background scans were recorded with a gold-coated integrating sphere.

## 2.4. Principal Component Analysis (PCA)

In NIR spectroscopy, there is a need for data-reduction methods because of the high number of spectral variables. PCA is used to decompose the matrix of the interest into several independent and orthogonal principal components [36]. An important application of PCA is classification and pattern recognition. The fundamental idea behind this approach is that data vectors representing objects in a high-dimensional space can be efficiently projected into a low-dimensional space by PCA and viewed graphically as scatter plots of PC scores. Objects that are similar to each other will tend to cluster in the score plots, whereas objects that are dissimilar will tend to be far apart. By "efficient," we mean the PCA model must capture a large fraction of the variance in the data set, say 70% or more, in the first few principal components [37]. Multiplicative Scatter Correction (MSC), a widely used pre-processing technique for NIRS was used to reduce the impact of light scattering. The concept behind MSC is that artifacts or imperfections (e.g., undesirable scatter effect) will be removed from the data matrix prior to data modeling. MSC is comprised of the following two steps:

1. Estimation of the correction coefficients (additive and multiplicative contributions),

$$x_{org} = b_0 + b_{ref,1} \cdot x_{ref} + e;$$

2. Correcting the recorded spectrum,

$$x_{corr} = \frac{x_{org} - b_0}{b_{ref,1}} = x_{ref} + \frac{e}{b_{ref,1}},$$

where $x_{org}$ is one original sample spectra measured by the NIR instrument, $x_{ref}$ is a reference spectrum used for pre-processing of the entire dataset, e is the un-modeled part of $x_{org}$, $x_{corr}$ is the corrected spectra, and $b_0$ and $b_{ref,1}$ are scalar parameters, which differ for each sample. In most applications, the average spectrum of the calibration set is used as the reference spectrum [38]. PCA and MSC were performed using Unscrambler X 10.4 (ver. 10.4, CAMO Software AS, Oslo, Viken, Norway, 2016) software.

## 2.5. Agglomerative Hierarchical Clustering (AHC)

Agglomerative Hierarchical Clustering (AHC), an unsupervised classification method, was used to group the coffee samples based on the obtained DFA coefficients (see Section 2.7). Ward method was used to compute the distance matrix and while the grouping was done using single linkage. Silhouette index was computed to obtain the optimal number of clusters. AHC and Silhouette index was computed using R-project (R version 4.0.0 (Arbor Day), R Core Team, Vienna, Austria, 2019) [39].

## 2.6. Multiple Correspondence Analysis (MCA)

Multiple Correspondence Analysis (MCA) is a multivariate method that provides a graphical representation of cross tabulations. Cross tabulations arise whenever it is possible to place events into two or more different sets of categories. CA is conceptually similar to principal component analysis but applies to categorical rather than continuous data. In a similar manner to principal component analysis, it provides a means of displaying or summarizing a set of data in two-dimensional graphical form [40]. MCA was carried out using XL-Stat software (ver. 2019.4.2, Addisonsoft, Paris, IDF, France, 2019).

### 2.7. Detrended Fluctuation Analysis (DFA)

Generally, DFA is used to analyze long-range correlations in time series. In case of NIRS, there is no time-related information; however, absorbance data as a function of wavenumber can be analyzed using DFA because of the large number of variables in the spectral data. The absorbance data are collected at regular interval depending on the resolution of the measurement. Detrended fluctuation analysis is a scaling analysis method providing a simple quantitative parameter—the scaling exponent $\alpha$—to represent the correlation properties of a signal [41]. The basic principle of DFA is to divide the time series into equal ranges and determine the fluctuation of the individual ranges by computing the scaling exponent. In order to create overlapping ranges, sliding window DFA approach was developed, which has been successfully used for pattern recognition and grouping. The presented paper uses the algorithm introduced by Radványi and co-workers [28]. Physisonet [42] was used to perform DFA and additional computations were carried out using R-project (R version 4.0.0 (Arbor Day), R Core Team, Vienna, Austria, 2019) [39].

### 2.8. Yield Stability Index

Yield stability index (YSI) was developed by Vízvári and Bacsi [34] and is able to capture the level of stability for a time series (or a signal) by fitting a trend line and quantifying the proportion of difference from the trend compared to a normal distribution as follows (Figure 1):

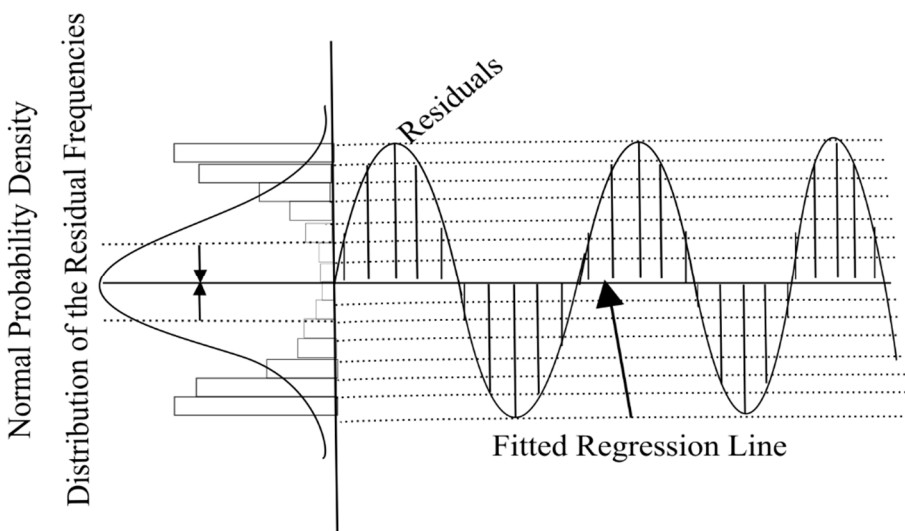

**Figure 1.** Visual representation of the underlying concept of yield stability index. Source: Papp and co-workers [43].

According to Vízvári and Bacsi [34], the signal should be normalized by dividing the whole signal by its mean and after that a trend line is fitted to the normalized signal (a polynomial of a degree of 1, 2 or 3) and the residuals are determined by subtracting the trend from the normalized signal. This process must be performed for all the investigated signals of a kind creating a single residual distribution function. We then fit a normal distribution and a histogram to the residual distribution. The favorable residual frequency (FRF) is the fraction of residuals falling into the middle four segments of the histogram and the favorable normal frequency (FNF) is the fraction of the middle 4 segments from the fitted normal distribution. YSI is calculated as the difference of the FRF and the FNF as follows:

$$\text{YSI} = 2 \times (\text{FRF} - \text{FNF})$$

The YSI index values should fall between −2 and 2, because both FRF and FNF can take values between 0 and 1. Negative values indicate that the distribution of the residuals significantly deviates

from normal implying larger oscillations around the trend line. In case the signal is stable, residuals should be located in the middle four segments around zero; therefore, the YSI should be zero or greater suggesting the lack of large oscillations.

We adapted and extended the methodology of Vizvari and Bacsi [34]. Our development compared to the above presented methodology is that we bin the values into [log2(n)] + 1 equal segments according to the Sturges' formula and further divided the middle four segments to 20 parts and we calculated the YSI value by gradually narrowing the border (dashed lines in Figure 1) of the middle four segments on both sides closer to 0 in 10 steps resulting 10 different YSI values. In the next step, we calculated how many times the YSI values were greater than 0 or a predefined cut value. In this way, we obtained a final index (between 0 and 10) that can be used to compare signal oscillation and stability—0 means higher oscillations and less stability and the signals were under the cut value most of the time, the higher values indicate stronger stability and less oscillations. Calculation of YSI were done using R-project (R version 4.0.0 (Arbor Day), R Core Team, Vienna, Austria, 2019) [39]. The signals were pre-processed by taking their logarithm to smooth the signals for the trend fit as much as possible.

## 3. Results

### 3.1. Spectral Characteristics

Before the statistical evaluation, it is practical to analyze the original spectra of the samples and first or second derivative of the spectra. First derivatives remove additive baseline shift so this is very useful in NIR spectroscopy. However, first derivatives produce peaks where the original spectrum had maximum slope and crosses zero where the original had a peak and are thus rather difficult to interpret. NIR spectra also tend to have linear baseline increases and these are removed by second derivatives which have negative peaks where the original had a peak and are thus more readily comprehensible. For these reasons second derivatives are often preferred [44]. Figure 2a shows that the spectral characteristics of raw and roasted coffee samples are similar in the main absorption bands, but the absorbance intensity of raw coffee is higher from 8500 to 3800 $cm^{-1}$. The differences evolve with the roasting process causing changes in chemical composition. The broad absorption bands (5210–5155 and 6945–6805 $cm^{-1}$) are assigned to water; however, water present in the sample affects the absorption of all other components.

The absorbance values of the sample spectra decreased in the spectral range 7500–3800 $cm^{-1}$, as roasting time increased (water content decreased), which are in accordance with earlier studies [14,21]. This characteristic was reversed in the spectral region nearest to visible range (12,500–8000 $cm^{-1}$). This phenomenon can be the result of the darker color of samples.

The second derivative spectra of coffee samples (see Figure 2b) are more detailed, because the overlapping peaks are resolved using the Savitzky–Golay transformation. The spectral characteristics of raw coffee samples can be discriminated visually from the roasted coffee spectra. During roasting, a large number of chemical transformations take place simultaneously (i.e., Maillard and Strecker reactions, degradation of proteins, polysaccharides, trigonelline and chlorogenic acids), leading to the formation of thousands of molecules that give the peculiar aroma of coffee [16]. The molecular overtone and combination bands seen in the NIR range are typically very broad, leading to complex spectra, thus, making it difficult to assign particular features to specific chemical components [6]. According to the scientific literature [14,17,21,23], a great amount of absorption bands are assigned to different chemical components which are perceptible on the second derivative spectra of coffee samples.

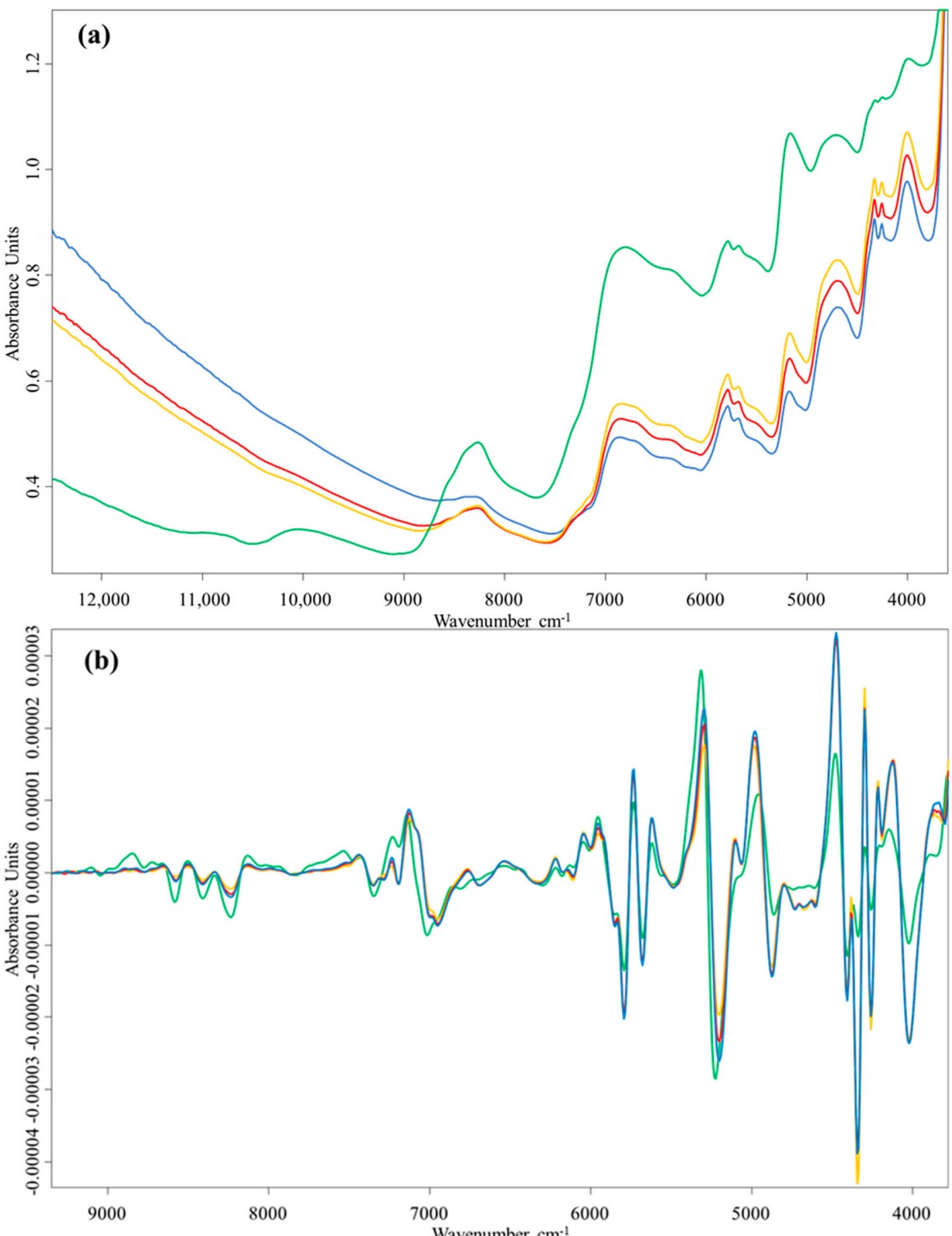

**Figure 2.** (**a**) The original average spectra of sample 1: raw (●), light roasted (●), medium roasted (●), dark roasted (●). (**b**) The second derivative spectra of sample 1: green (●), light roasted (●), medium roasted (●), dark roasted (●).

## 3.2. Principal Component Analysis (PCA)

PCA was performed on the original and transformed (MSC) data in the 12,500–3800 cm$^{-1}$ spectral range. In case of the original data, the first principal component had a smaller impact on the separation of the samples. There is a tendency among the samples, namely the dark roasted samples are

characterized by negative scores along PC1, while medium and light roasted samples are located to the right of side of the plot (plot not shown). MSC transformation was performed on the spectra of each roasting stage separately.

Figure 3a shows the scores plot of the first two principal components (PCs), which capture 98% of the variance. The main direction is described by PC1 (93%). According to the loading plot, the variables between 12,500–7500 cm$^{-1}$ have a negative effect on the sample distribution along PC1, therefore all dark roasted samples can be found on the left side of scores plot. These features have a high impact on differentiating roasting stages. Both PC1 and PC2 values play a role in the separation of the roasting levels. It has to be noted that the loading plot values of PC3 (1%) describes important absorption bands (of the 6800–3800 cm$^{-1}$ region; see in Section 3.3.).

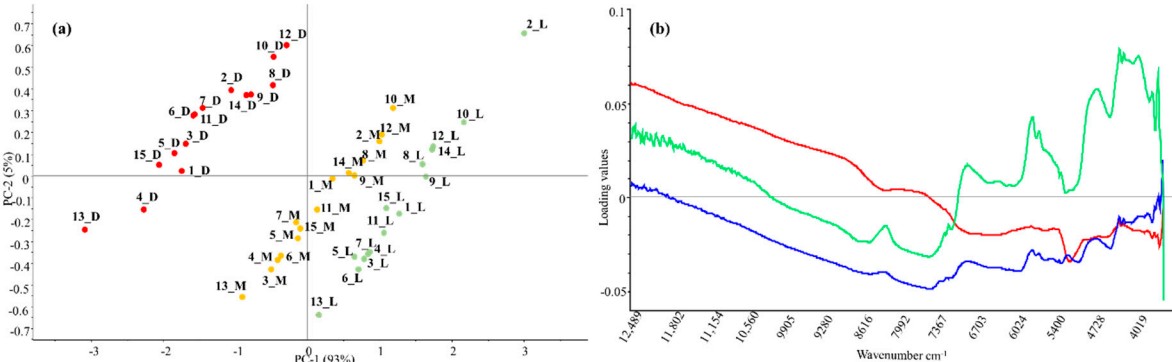

**Figure 3.** (**a**) Scores plot (PC1 vs PC2) of roasted coffee samples labelled with sample number and roasting level (L—light; M—medium; D—dark roasted). (**b**) Loading plots of the first three PC: PC1 (•), PC2 (•), PC3 (•).

### 3.3. Detrended Fluctuation Analysis

Detrended fluctuation analysis (DFA) was run on the average FT-NIR spectra of six parallel measurements of the same sample. The obtained $\alpha_1$ and $\alpha_2$ coefficients were used to plot the samples and to compute their distances. DFA differentiated green samples from the roasted samples clearly. As Figure 4 shows, samples having an $\alpha_1 = 1.95$ and $\alpha_2 = 1.65$ coefficients can be classified as green samples. Differentiation among green samples is done by the distinct chemical composition, which are also affected by diverse cultivation and post-harvest processes. The size, shape and color of the beans, as well as the silver skin and parchment left on their surface, also have an effect on the spectra.

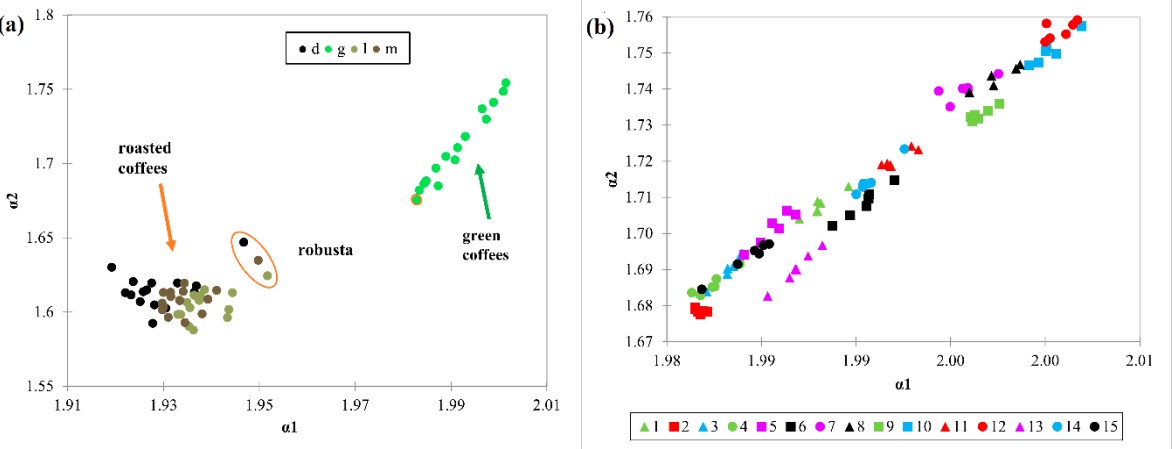

**Figure 4.** $\alpha_1$ and $\alpha_2$ coefficients obtained from detrended fluctuation analysis. (**a**) averaged spectra of the green (g), light (l), medium (m) and dark roast (d) samples. Different colors denote different roasting levels (**b**) the six replicates of the green samples. Different colors and symbols denote different countries.

According to the second derivative spectra of the green samples, smaller but noticeable differences can be observed. These are manifested in band shifts and higher or smaller absorption for the given band location, which are assigned for example to water (5210 cm$^{-1}$), fatty acids (around 5600–4000 cm$^{-1}$, 8300–8100 cm$^{-1}$), fibers (around 7450 cm$^{-1}$, 5800–4700 cm$^{-1}$), caffeine (around 8665, 8250, 5300–5100 and 4700 cm$^{-1}$) and proteins (5700, 4800–4600 cm$^{-1}$).

The three roasting levels show a distinct pattern, where the dark roasted samples usually show lower $\alpha_1$ and higher $\alpha_2$ coefficient values. Light roasted samples are grouped on the bottom right corner of the plot, having the highest $\alpha_1$ and lowest $\alpha_2$ coefficient values. Based on these, a straight line can be drawn which shows the roasting level of the individual samples (Figure 5).

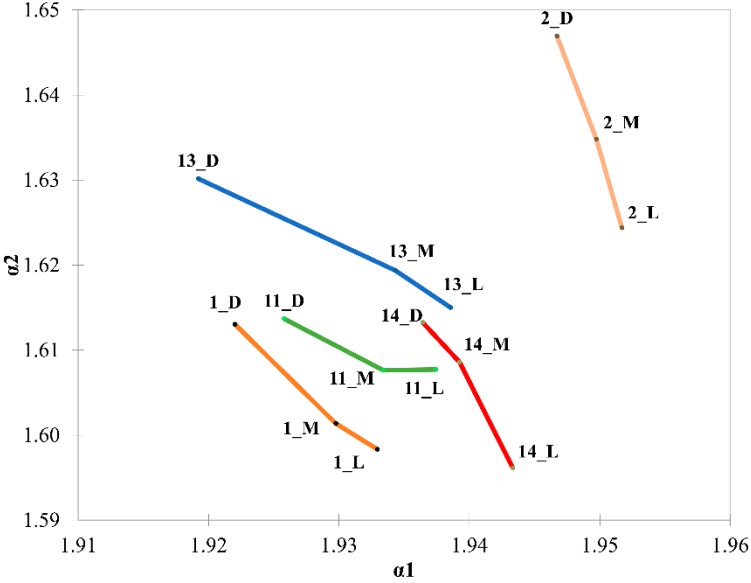

**Figure 5.** $\alpha_1$ and $\alpha_2$ coefficients obtained from detrended fluctuation analysis. Averaged spectra of the green, light, medium and dark roast samples. Different colors denote different coffee origins.

When the DFA coefficients are evaluated within one sample, the differentiation among the roasting levels are clear. Figure 6 introduces the $\alpha_1$ and $\alpha_2$ coefficients of the six parallel measurements of the Columbian sample (all roasting levels). Although it is visually clear that the samples do not overlap, we used Agglomerative Hierarchical Clustering to justify the grouping of the samples. On the right of Figure 6, we can see the results of AHC. Clear clusters are formed, no misclassification was done. It can also be seen that medium and light roasted samples are located closer to each other than the dark roasted samples.

There is some differentiation among the cultivars, the only robusta sample is placed further from the other arabica samples. It is caused by the distinct spectral features of the robusta sample, since robusta coffees contain diverse levels of caffeine, chlorogenic acids, lipids and sucrose compared to arabica coffees [5]. It has to be noted, however, that we analyzed only one robusta sample, hence further generalization cannot be drawn. The three roasting levels cannot be differentiated clearly when all samples are analyzed together, however, within one sample the roasting levels were differentiated.

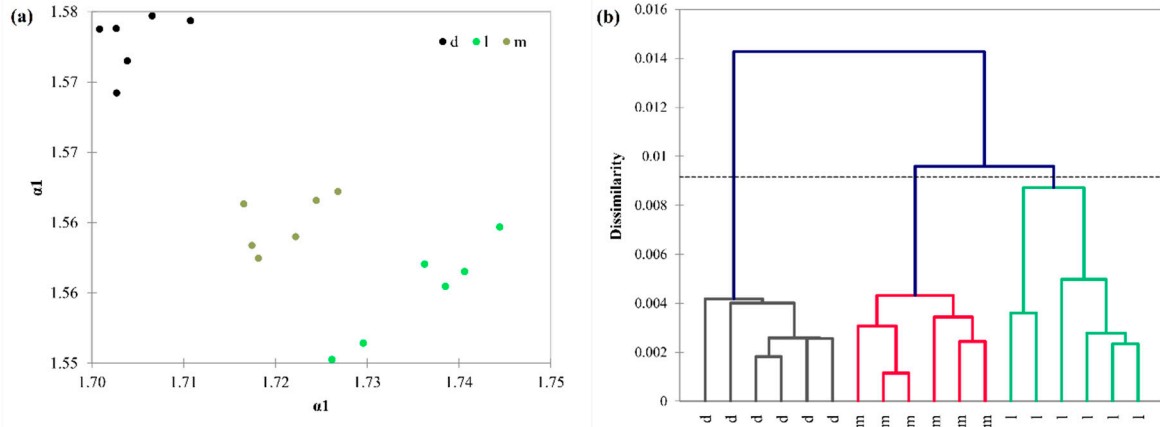

**Figure 6.** Detrended fluctuation analysis of the parallel measurements of the Columbian coffee samples. (**a**) $\alpha_1$ and $\alpha_2$ coefficients obtained from detrended fluctuation analysis. Spectra of the light (l), medium (m) and dark (d) roast samples are coded by different colors. (**b**) Agglomerative hierarchical cluster analysis of the $\alpha_1$ and $\alpha_2$ coefficients using Euclidean distance and complete linkage.

### 3.4. Yield Stability Index

Yield stability index gives an additional information about the analyzed spectra. As YSI attempts to filter extremities among the set of the analyzed spectra, the obtained result of the YSI analysis is a contingency table, where the sample sets (in our case the roasting levels such as green, light, medium and dark) are presented, while in the rows the number of times the YSI index was over the cut value. Table 2 presents the numerical results obtained after running the YSI on the original NIR data set. A chi-square test indicated that the rows and columns of the table shows significant associations ($\chi^2$ (24, N = 36) = 49.31, p = 0.0017). In order to visualize the observed associations, correspondence analysis was run on the contingency table presented by Table 2. Correspondence analysis visualized the similarities among the green and light roasted samples (Figure 7). These samples show similar distribution of residuals, close to the mean but from the cut values we can see that light roast is more stable as its signal had higher YSI values. As the roasting time increases, the difference from the mean trend also increases, meaning that dark roasted samples show extreme fluctuations.

**Table 2.** Distribution of YSI indices by roasting level *.

| YSI*n* | Green | Light Roast | Medium Roast | Dark Roast |
|---|---|---|---|---|
| 0 | 0 | 0 | 0 | 0 |
| 1 | 0 | 0 | 0 | 0 |
| 2 | 0 | 2 | 0 | 1 |
| 3 | 1 | 1 | 4 | 0 |
| 4 | 1 | 4 | 3 | 3 |
| 5 | 5 | 5 | 3 | 0 |
| 6 | 2 | 2 | 1 | 0 |
| 7 | 1 | 1 | 4 | 0 |
| 8 | 1 | 0 | 0 | 0 |
| 9 | 2 | 0 | 0 | 6 |
| 10 | 2 | 0 | 0 | 5 |
| *N* | 15 | 15 | 15 | 15 |
| Cut value for YSI | 0 | 0.2 | 0.1 | 0 |

* quadratic polynomial was used in case of all signals except green roast where a linear polynomial fitted the best.

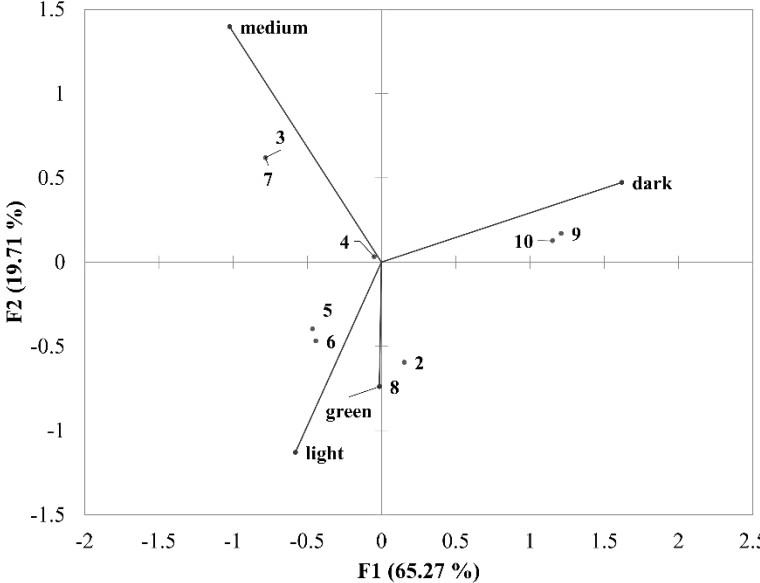

**Figure 7.** Correspondence analysis run on the data presented by Table 2. Numbers denote the number of times YSI was over the cut value.

Table 2 presents the results of yield stability index (YSI) analysis. The number of spectra over the YSI cut value is presented in each row. Grey background indicates the highest numbers of occurrences. *N* denotes sample number. YSI*n*: Number of times YSI was over the cut value.

## 4. Conclusions

Differentiation of the roasting levels was successfully completed using principal component analysis (PCA), however, only after applying multiple scatter correction (MSC). MSC had no effect on the differentiations when detrended fluctuation analysis (DFA) was applied. It has to be noted that PCA was able to differentiate the different roasting levels when all samples were analyzed together. The three roasting levels cannot be differentiated clearly by DFA as only the green samples were differentiated clearly and the three roasting levels showed only tendencies. On the other hand, DFA successfully differentiated the roasting levels sample-wise, meaning that it captured the within-sample differences well. It is an important aspect here since usually the question of producers is how to tell if a green coffee is roasted well (e.g., on a level that highlights the proper quality of the given coffee). A simple cluster analysis applied on the coefficients of the DFA showed clear groupings, making the results robust and creating the opportunity to perform classifications as well. Another advantage of DFA here comes from the nature of the method. DFA analyses one spectrum at a time, therefore it is not influenced by the others. When performing PCA, the whole data set (e.g., all samples) are analyzed at the same time. Naturally, the more samples we have the better results we can get in terms of validity and generalization. However, it is a disadvantage when we have a set of new samples since the scores obtained from PCA will change. Using DFA, global thresholds can be set for each sample, e.g., if the $\alpha_1$ and $\alpha_2$ coefficients are within the given ranges, the sample can be classified as green, light, medium or dark roasted. Therefore, DFA might be a valuable tool in coffee quality control. It is also an advantage, when the number of samples (or spectra) is too low to perform multivariate data analyses. Yield Stability Index was first used in signal processing. The methodology was able to detect that light roast is the most stable among all roasting levels.

Future research should focus on the application of DFA in terms of the analysis of the effects of other transformation methods of the spectra. Additionally, different samples should also be analyzed to evaluate the robustness of the method, e.g., if there are any effects of the samples being analyzed on the performance of DFA.

**Author Contributions:** E.B.: Data curation, Formal analysis, Investigation, Validation, Visualization, Writing—original draft, Writing—review & editing; M.F.: Data curation, Validation, Supervision, Project administration, Resources, Writing—original draft, Writing—review & editing; S.K.: Formal analysis, Methodology, Software, Resources, Supervision, Visualization, Writing—original draft, Writing—review & editing; A.G.: Formal analysis, Funding acquisition, Methodology, Project administration, Supervision, Writing—original draft, Writing—review & editing. All authors have read and agreed to the published version of the manuscript.

**Funding:** The Project was funded by the European Union and co-financed by the European Social Fund (grant agreement no. EFOP-3.6.3-VEKOP-16-2017-00005). The APC was funded by the European Union and co-financed by the European Social Fund (grant agreement no. EFOP-3.6.3-VEKOP-16-2017-00005).

**Acknowledgments:** A.G. thanks the support of the Premium Postdoctoral Researcher Program of the Hungarian Academy of Sciences. This project was supported by the János Bolyai Research Scholarship of the Hungarian Academy of Sciences. The Project is also supported by the Doctoral School of Food Science SZIU.

**Conflicts of Interest:** The authors declare no conflict of interest.

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
