# Peer review of "Application of Detrended Fluctuation Analysis and Yield Stability Index to Evaluate Near Infrared Spectra of Green and Roasted Coffee Samples"

_processes, doi:10.3390/pr8080913_

Round 1

Reviewer 1 Report

Dear authors, your manuscript (processes-862634) describes new techniques for data analysis of near infrared spectra of green and roasted coffee samples and discusses the advantages and short comes of the two methods. The manuscript is well written and most of the time comprehensible. However, I have several suggestions to improve the manuscript as described in detail in the attached document.

Author Response

Reviewer 1:

First, we would like to thank your work to help to make our manuscript better. We did our best to improve the manuscript according your observations and to answer all of your questions. Our answers are listed below.

Abstract

Line 23: were instead of was analyzed

Response: The sentence has been corrected: “Fourteen green coffee samples from all over the world have been roasted on three different levels and their NIR spectra were analyzed.”

Introduction

I would shorten the introduction and focus on the topics of your study, the roasting process and its stability.

Skip line 37 – The bulk… - to line 44 - … roasted coffee beans. – Blending and food authentication is not the topic of your presented research.

If you keep this part: ref 3: please include more and more recent publications.

Response: Thank you for remark. The part in question has been deleted from the manuscript.

Skip line 45 to line 53. If not:

Response: The comment is not clear here. If the reviewer means to delete the paragraph, we would add that this paragraph intended to introduce the high variability of coffee varieties, making their classification a rather difficult task. Our presented method aims to help in this complex task.

Line 46: Pierre?

Response: Coffea canephora Pierre is the correct denomination of robusta plant.

Gunning, Y.; Defernez, M.; Watson, A.D.; Beadman, N.; Colquhoun, I.J.; Le Gall, G.; Philo, M.; Garwood, H.; Williamson, D.; Davis, A.P.; et al. 16-O-methylcafestol is present in ground roast Arabica coffees: Implications for authenticity testing. Food Chem. 2018, 248, 52–60.

Cheng, B.; Furtado, A.; Smyth, H.E.; Henry, R.J. Influence of genotype and environment on coffee quality. Trends Food Sci. Technol. 2016, 57, 20–30.

Line 53: ref 7 does not fit

Response: We state that “It  (NIR technology) is widely used in different fields of food industry for qualitative and quantitative analysis of products from raw material to finished products” Ref 7 introduces the application of NIR on snack prodcuts, which are finished products.

Line 56 and 62: ref 8: please include more references

Response: References were added to the section accordingly:

Workman, J.; Weyer, L. Practical Guide to Interpretive Near‐Infrared Spectroscopy; CRC Press: Boca Raton, 2008; ISBN 9780429119576.

Pasquini, C. Near infrared spectroscopy: Fundamentals, practical aspects and analytical applications. J. Braz. Chem. Soc. 2003, 14, 198–219.

Line 61: “cooked through” I would change this expression. Cooking is a watery process, roasting is a dry one.

Response: The expression has been changed according to the suggestion.

“The beans are properly roasted and all the potential details of the coffee are apparent.”

Line 63: … is a nondestructive …

Line 64: … with a minimal …

Response: The grammar mistakes have been corrected.

Line 71: …roasting degree … ref 16: are there more recent studies?

Response: To the best of our knowledge, no other study is available on this particular topic.

Line 79-83: which multivariate statistical studies did the studies mentioned in line 68-79?

Response: The manuscript has been supplemented accordingly.

“In the afore mentioned scientific papers, different pre-processing techniques (MSC, SNV, 1st and 2nd derivatives, normalization, OSC) and multivariate statistical tools such as PCA, PLSR, linear discriminant analysis (LDA) and PLS-DA were used to analyze various properties of coffee.”

Materials and Methods

Line 116: three parts …

Response: The grammar has been corrected.

Table 1: In coffee business, “Growing region” is the region within one country where the coffee grows, or even on a farm level, but not the continent of the growing country. Please skip it.

Response: Thank you for the consideration. Table 1 has been modified accordingly.

Line 126-128: what was the overall roasting time?

Response: The roasting time varies from sample to sample, as the time of the first crack is related to the composition of the given coffee.

Line 138: so you did experiments on the whole beans, not the ground powder, right?

Response: Yes, we did our experiment on the whole beans.

Paragraph 2.4 PCA: PCA is a rather well known method, please skip the detailed description of the method itself and focus on your data treatment. Did you do any data pre-treatment except msc, for example?

Response: The description of PCA has been curtailed and more detailed has been added to the manuscript.

“Multiplicative Scatter Correction (MSC), a widely used pre-processing technique for NIRS was used to reduce the impact of light scattering. The concept behind MSC is that artifacts or imperfections (e.g., undesirable scatter effect) will be removed from the data matrix prior to data modeling. MSC comprises two steps:

  1. Estimation of the correction coefficients (additive and multiplicative contributions).

                                                                                       (

  1. Correcting the recorded spectrum:                                                                     

where xorg is one original sample spectra measured by the NIR instrument, xref is a reference spectrum used for pre-processing of the entire dataset, e is the un-modeled part of xorg, xcorr is the corrected spectra, and b0 and bref,1 are scalar parameters, which differ for each sample. In most applications, the average spectrum of the calibration set is used as the reference spectrum.”

Did you do any data pre-treatment except msc, for example?

Response: 1st  and 2nd derivative and SNV pre-treatments were also used, but their results were not satisfactory.

Paragraph 2.5 DFA: any data pre-treatment?

Response: DFA was run after MSC.

General: Please add agglomerative hierarchical cluster analysis (Fig 7) and correcpondence analysis (Fig 8).

Response: The two methods have been added into the methods section.

Results

Line 217-219: some words are missing in this sentence.

Response: The sentence has been rephrashed as follows:

“The broad absorption bands (5210 – 5155 cm-1 and 6945 – 6805 cm-1) are assigned to water, however water present in the sample affects the absorption of all other components.”

Line 220-222: Please clarify this sentence.

Response: The sentence has been clarified as “The absorbance values of the sample spectra decreased in the spectral range 7500 – 3800 cm-1, as roasting time increased (water content decreased), which are in accordance with earlier studies [14,21].”

Line 231: “The second derivative…more detailed (see Fig 3), …”

Response: The indication of the Figure has been added to the manuscript.

Line 232-233: In my opinion, the differences between green and roasted samples was clearer within the original average spectra (Fig 2).

Response: You have on the one hand right, however the higher water content of green coffee sample causes increase in the baseline. The smaller and more significant differences can be seen on the 2nd derivative spectra of samples.

Line 236: ref 18: Please include more references.

Response: References have been added.

Farah, A. Coffee: Production, Quality and Chemistry; Royal Society of Chemistry, 2019; ISBN 9781782622437.

Hu, G.; Peng, X.; Gao, Y.; Huang, Y.; Li, X.; Su, H.; Qiu, M. Effect of roasting degree of coffee beans on sensory evaluation: Research from the perspective of major chemical ingredients. Food Chem. 2020, 331.

Line 240-241: Can you give an example of assigned absorption bands?

Response: Band assingments are included in section 3.3.

Line 250: “According to the loading plot…”

Response: The grammar has been corrected.

Line 250: Could you include the figure of the loading plot?

Response: The figure of loading plot (Fig 4b) has been added.

Line 254-255: “…important absorption bands…”: important for what?

Response: The specific absorption bands are included in section 3.3. It has been marked in the end of the sentence.

“It has to be noted, that the loading plot values of PC3 (1%) describes important absorption bands (of the 6800 – 3800 cm-1 region; see in section 3.3.).”

General: in line 213-214, you mention original spectra, 1st and 2nd derivative, but you do not show/discuss the 1st  derivate. Could you include it in at least on sentence and/or Figure?

Response: Section 3.1. has been replenished with additional information.

Line 248: different font size

Response: Font size has been standardized.

Fig 4: Where is the green coffee situated?

Response: Fig 4 shows the scores plot of roasted coffee samples. The caption has been replenished for better understanding.

Paragraph 3.3 DFA: You mention the average FT-NIR spectra of one sample in the first paragraph and the 2nd deriv ative in the second paragraph. Could you please clarify which data set you analyzed with DFA with (a) average spectra and (b) with 2nd derivative? And for which analysis did you put all samples and roasting levels together, and for which analysis you just took one sample?

Response: 2nd derivatives of the parallel measurements have been averaged and used for input by DFA.

Fig 5: Remark: It is interesting how different green coffees are even with the same origin or from the same continent. Maybe you could use/discuss this to differentiate origins.

Response: Thank you for your comments. We plan to move on to this direction, however, we should conduct further analytical measurements to see which parameters are responsible for the differentiation.

Line 281-282: Please refer to Fig 6 at some point.

Response: Refence to the figure has been added.

Line 289: “The three roasting levels cannot be differentiated clearly.” Between all samples this is true, but within one coffee sample, you can differentiate.

Response: Thank you for this comment, we added it.

Fig 6: Caption: “Different colors denote different roasting levels coffee origins.”

Response: The caption has been corrected accordingly.

Fig 6: Where is the green coffee situated?

Response:

Fig 7: Discussion of this Fig is missing!

Response: Discussion has been added.

Fig 7: Caption: “…green (g)…” green coffee samples are not included in this Fig!

Response: The caption has been corrected accordingly.

Line 303: “to filter identify”: words are missing or too many words…

Response: The word “identify” has been removed.

Line 306 following and line 315 following: repetition. Please merge.

Response: The repetition has been corrected.

Conslusions

Line 328: “analysis (DFA).” Something is missing, like “was performed”.

Response: The grammar has been corrected.

Line 329: “… were analyzesd together.”

Response: The grammar has been corrected.

Line 343-344: What is the limit of number of samples for DFA?

Response: There is no limit, since DFA takes one spectrum at a time and evaluates it, regardless of the other spectra. It means that the limit of number of samples could only come from the used computer’s hardware.

Line 344-345: What do we learn from that?

Response: The following sentence has been inserted here: Therefore, DFA might be a valuable tool in coffee quality control.

Reviewer 2 Report

In this manuscript Benes et al, are using innovative and powerful methodology of detrended fluctuation analysis (DFA) and yield stability index (YSI) to analyze NIR spectra of different roasting level of coffee. The authors were able to identify roasted samples and differentiate roasting level using DFA. Through YSI the authors concluded light roast as most stable one. Overall, the article is of general interest to the audience of Processes.

Recommendation: Publish with minor changes

Minor Points:

  1. In almost all figures, the axis is hard to read. It is suggested that authors increase the font size of the figures.
  2. Page 11, Line 328. It seems the sentence is incomplete.
  3. Line 329: It should be analyzed.
  4. Line 330 – 331. The sentence needs to be restructures for clarity.

Author Response

Reviewer 2:

First, we would like to thank your work to help to make our manuscript better. We did our best to improve the manuscript according your observations and to answer all of your questions. Our answers are listed below.

In almost all figures, the axis is hard to read. It is suggested that authors increase the font size of the figures.

Response: The figures have been revised and grouped. Necessary changes have been made where needed (especially in case of figure 3 and 5). The important information can now be better seen on all figures.

Page 11, Line 328. It seems the sentence is incomplete.

Response: Thank you for noticing the mistake, the sentence was finished as: “MSC had no effect on the differentiations when detrended fluctuation analysis (DFA) was applied”.

Line 329: It should be analyzed.

Response: The correction has been made according to this: “when all samples were analyzed together”.

Line 330 – 331. The sentence needs to be restructures for clarity.

Response:

The sentence was restructured as: “The three roasting levels cannot be differentiated clearly by DFA as only the green samples were differentiated clearly and the three roasting levels showed only tendencies”.
